# Prior-Free Arterial Spin Labeling Quantification via 3D Physics-Informed Neural Networks

**Alessandro Giupponi**[1] (iD)                           ALESSANDRO.GIUPPONI@PHD.UNIPD.IT

**Chiara Da Villa**[1]                                    CHIARA.DAVILLA@STUDENTI.UNIPD.IT

**Giulio Ferrazzi**[2] (iD)                               GIULIO.FERRAZZI@PHILIPS.COM

**Francesca Benedetta Pizzini**[3] (iD)        FRANCESCABENEDETTA.PIZZINI@UNIVR.IT

**Matthias J.P. Van Osch**[4] (iD)                        M.J.P.VAN_OSCH@LUMC.NL

**Mattia Veronese**[1,5] (iD)                             MATTIA.VERONESE@UNIPD.IT

**Marco Castellaro**[1] (iD)                              MARCO.CASTELLARO@UNIPD.IT

[1] *Department of Information Engineering, University of Padova, Padova, Italy*

[2] *Philips Healthcare, Milan, Italy*

[3] *Department of Engineering for Innovation Medicine, University of Verona, Verona, Italy*

[4] *C.J. Gorter MRI Center, Department of Radiology, Leiden University Medical Center, Leiden, Netherlands*

[5] *Neuroimaging Department, IoPPN, King's College London, London, UK*

**Editors:** Under Review for MIDL 2026

## Abstract

Arterial Spin Labeling parameter estimation is a non-linear inverse problem limited by low signal-to-noise ratio (SNR) and complex parameter dependencies. Traditional Variational Bayesian (VB) methods mitigate this via explicit spatial priors, which can bias results in low-SNR regions. We propose a 3D Physics-Informed Neural Network (PINN) treating perfusion quantification as a continuous Implicit Neural Representation. By embedding the Buxton model as a differentiable constraint, the framework maps spatio-temporal coordinates to perfusion parameters. While avoiding hard-tuned spatial smoothing, the architecture leverages implicit architectural priors via multi-resolution hash-encoding to capture anatomical details and overcome the spectral bias of standard Multi Layer Perpetrons. Evaluated on 18 healthy subjects, the PINN demonstrated superior data consistency and anatomical contrast compared to state-of-the-art VB estimators, offering a physiology grounded alternative for high-fidelity 3D perfusion quantification.

**Keywords:** Arterial Spin Labeling, Physics-Informed Neural Network, Hadamard Encoding, Cerebral perfusion.

## 1. Introduction

Arterial Spin Labeling (ASL) MRI allows for non-invasive perfusion quantification, yet its utility is limited by low signal-to-noise ratio (SNR) and a non-linear dependence on parameters like arterial transit time (ATT). Time-encoded pseudo-continuous ASL (te-pCASL) improves efficiency through Hadamard encoding, enabling simultaneous cerebral blood flow (CBF) and ATT estimation (Woods et al., 2024). Conventional quantification typically relies on Variational Bayesian (VB) estimators (Chappell et al., 2009), which stabilize the inversion through explicit spatial priors that may introduce bias in low-SNR regions. To address this, we present a 3D Physics-Informed Neural Network (PINN) treating perfusion quantification as an Implicit Neural Representation (INR) problem. By embedding

the Buxton model (Buxton et al., 1998) as a differentiable constraint, the network replaces explicit Gaussian priors with implicit architectural regularization, leveraging coordinate-based learning and multi-resolution hash encoding to learn a continuous representation of the perfusion volume and maintain anatomical fidelity (de Vries et al., 2023; de Vries et al., 2024).

## 2. Materials and Methods

**Data.** Single-echo te-pCASL data (11 sub-boli, total labeling = 3600 ms, PLD = 49 ms, resolution = $3 \times 3 \times 7 \ mm^3$) from 18 healthy volunteers were motion-corrected, Hadamard decoded (Gunther, 2007), and normalized to the blood equilibrium magnetization ($M_{0b}$).

**PINN Framework.** Our architecture utilizes two SIREN-based (Sitzmann et al., 2020) Multi-Layer Perceptrons (MLPs), $f_{tissue}$ and $f_{ode}$. While $f_{tissue}(x, y, z, t)$ (3 layers, 16 units) maps spatio-temporal coordinates to the ASL signal and learns the 4D signal manifold, $f_{ode}(x, y, z)$ (3 layers, 16 units) maps spatial coordinates directly to CBF and ATT. To capture fine-grained spatial details, we apply multi-resolution hash encoding (Müller et al., 2022) to the input spatial coordinates, using 16 levels and 2 features per level, with a base resolution of 16 and a finest resolution of 4096, based on evidence coming from similar work (de Vries et al., 2024). This expands positional information into multi-resolution embeddings, improving spatial variability modeling while maintaining compact networks. The framework is optimized end-to-end via a multi-task loss function $L$:

$$L = \alpha \, ||s - f_{tissue}||_2^2 + \lambda \, ||\frac{\delta f_{tissue}}{\delta t} - P\left(f_{ode}, t\right)||_2^2$$

where $P$ is the differentiable Buxton operator, $\frac{\delta f_{tissue}}{\delta t}$ is the temporal derivative of the first network prediction computed via automatic differentiation, and $\alpha$ and $\lambda$ were set respectively to 1 and 0.5 for training stability and optimal performance. Temporal derivatives of the Buxton model were incorporated via a fifth-order spline that models the time-varying sub-boli durations. Optimization used AdamW (learning rate = $10^{-3}$, batch size = 1000, epochs = 5000) with cosine annealing. This formulation ensures that the learned signal manifold $f_{tissue}$ is physically consistent with the predicted parameters $f_{ode}$, with the ODE acting as a temporal regularizer, preventing the high-capacity hash encoding from overfitting stochastic noise.

**Baseline and Evaluation.** The 3D PINN was compared to a VB estimator (BASIL, (Chappell, 2023)) using a Gaussian ATT prior (mean/SD = 1s/1s). Performance was assessed via Root Mean Square Error (RMSE) of the residuals between the measured signals and the Buxton-model reconstruction using the estimated parameters. Quantitative evaluation in gray matter (GM) and white matter (WM) was paired with a qualitative assessment of spatial smoothness, noise robustness, and tissue contrast preservation.

## 3. Results and Discussion

Quantitative analysis (Tab. 1) confirms that the PINN consistently achieved higher fitting accuracy than VB, with an average RMSE reduction of 46% in GM and 37% in WM. Notably, this error reduction was consistent across the brain, with the PINN yielding lower RMSE in over 74% of GM voxels and 76% of WM voxels. While lower RMSE can sometimes indicate noise-fitting, validation on digital phantoms across different level of noise (detailed in Supplementary Material) confirms that the PINN-predicted signals remained

smooth and physically consistent and this stability translates to improved parameter accuracy and higher Structural Similarity Index Metrics (SSIM) and Pearson Correlation Coefficient (PCC) compared to regularized nonlinear estimators.

| Region | RMSE VB | RMSE PINN | improvement ratio |
|---|---|---|---|
| Gray Matter | $1.276 \pm 0.385$ | $\mathbf{0.691 \pm 0.178}$ | $1.946 \pm 0.810$ |
| White Matter | $0.900 \pm 0.171$ | $\mathbf{0.563 \pm 0.164}$ | $1.676 \pm 0.441$ |

Table 1: Quantitative comparison of fitting performance between VB and PINN across tissue types. The improvement ratio is defined as RMSE(VB)/RMSE(PINN).

Qualitatively, PINN maps exhibit sharper GM-WM contrast and reduced noise, particularly in the ATT maps (Fig. 1). While VB estimates often revert toward the prior mean in low-SNR regions (notably WM), the PINN leverages the spatial consistency of the 3D volume and the hash encoding to produce physiologically coherent parameter distributions. These findings suggest that coordinate-based neural networks, when constrained by physical laws, can effectively regularize the solution space via implicit architectural priors rather than explicit Gaussian spatial prior, maintaining high-frequency anatomical fidelity without prior-induced bias.

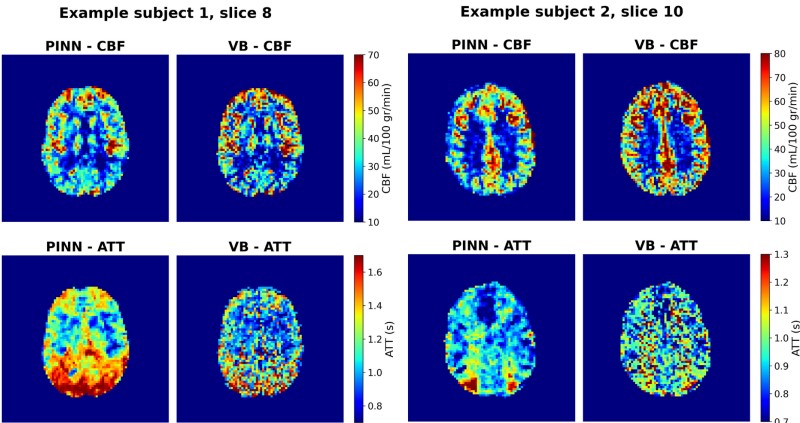

Figure 1: Qualitative comparison of CBF (top) and ATT (bottom) maps for two representative subjects and slices.

## 4. Conclusion

This work presents a 3D physics-informed framework for te-pCASL quantification. By integrating the Buxton model as a differentiable constraint within a coordinate-based INR, the PINN provides more accurate and anatomically consistent perfusion maps than conventional Bayesian methods. This approach offers a data-efficient and physically interpretable alternative for perfusion parameters estimation, but future work will involve testing the framework's generalizability across pathological conditions (e.g., stroke or dementia) where delayed ATTs and low-SNR provide a greater challenge for parameter estimation.

## Acknowledgments

Philips Healthcare support this research through a research agreement.

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

# Appendix A. Implementation details

## A.1. Configuration

Table 2 lists all the implementation details for the proposed two stage physics-informed framework.

## A.2. Accuracy Validation via Digital Phantom Simulations

To develop the framework and test its robustness to noise, we conducted a simulation study using the Boston ASL Template and Simulator (BATS) (Taso et al., 2022). Synthetic te-pCASL signals were generated with spatially heterogeneous ground-truth (GT) maps for CBF and ATT as well as equilibrium magnetization ($M_0$), resampled to match real data resolutions ($3 \times 3 \times 7$ mm$^3$).

We compared the PINN framework against a regularized Nonlinear Least Squares (NLLS) baseline across two temporal signal-to-noise ratio (tSNR) levels, 1.5 and 0.5. This is achieved by adding Gaussian noise following established methodologies (Bladt et al., 2020). The regularized NLLS estimates CBF and ATT voxel-wise by minimizing the residuals between the observed ASL signal and the model, with a small penalty on ATT deviations from its initial guess. Optimization was performed with soft-L1 loss and physiologically plausible bounds, using the Powell's hybrid method. As shown in Table 3 and Figure 2, the PINN outperformed the voxel-wise baseline in recovering spatial structure, particularly for ATT, which is traditionally more sensitive to noise.

| Configuration | Value |
|---|---|
| Number of layers | |
| $f_{tissue}(x,y,z,t)$ | 3 |
| $f_{ode}(x,y,z)$ | 3 |
| Neurons per layer | |
| $f_{tissue}(x,y,z,t)$ | 16 |
| $f_{ode}(x,y,z)$ | 16 |
| Activation function | |
| $f_{tissue}(x,y,z,t)$ | Siren $\omega = 15$, $\omega_0 = 15$ |
| $f_{ode}(x,y,z)$ | Siren $\omega = 5$, $\omega_0 = 25$ |
| $f_{ode}(x,y,z)$, last layer | Exponential |
| Optimizer | |
| Base learning rate | $10^{-3}$ |
| Learning rate schedule | Cosine Annealing |
| Annealing period ($T_{max}$) | 5000 |
| Final learning rate | $10^{-10}$ |
| Batch size | 1000 |
| Hash Encoding | |
| Number of levels | 16 |
| Features per level | 2 |
| $log_2$ hashmap size | 15 |
| Base resolution | 16 |
| Finest resolution | 4096 |
| GPU Memory requirement (MB) | 2521 |

Table 2: Implementation configuration for the two stage PINN framework proposed, including networks' hyperparameter, optimization choices and hash-encoding parameters.

| tSNR | | SSIM | | PCC | |
|---|---|---|---|---|---|
| | | PINN | NLLS | PINN | NLLS |
| 1.5 | CBF | **0.825** | 0.786 | **0.969** | 0.944 |
| | ATT | **0.951** | 0.467 | **0.995** | 0.761 |
| 0.5 | CBF | **0.643** | 0.533 | **0.907** | 0.841 |
| | ATT | **0.842** | 0.212 | **0.980** | 0.482 |

Table 3: SSIM and PCC obtained in CBF and ATT estimation by the two different approaches, with tSNR of 1.5 and 0.5.

## A.3. Hash-Encoding

The spatial encoding maps 3D coordinates into a high-dimensional feature space to overcome the spectral bias of standard MLPs, enabling the recovery of fine-grained anatomical details. Figure 2 illustrate this process in a 2D slice, assuming just two resolution ($L = 2$), while out methods utilizes $L = 16$ resolutions.

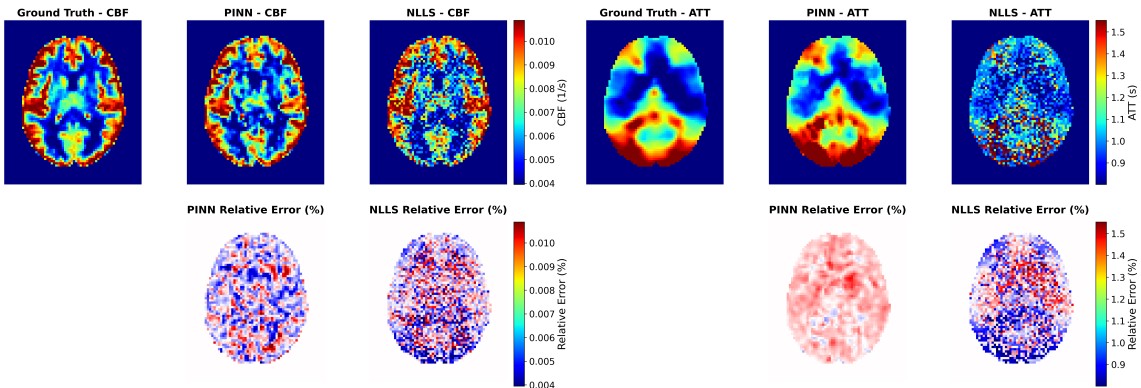

Figure 2: Comparison of CBF and ATT estimations using PINN and NLLS. The first row shows the estimated maps, while the second row presents the corresponding Percent Relative Error maps.

The spatial domain is partitioned into a hierarchy of $L$ nested grids with increasing resolutions. For any continuous coordinate $x \in \mathbb{R}^3$, the framework identifies the neighboring $2^3$ voxel vertices at each resolution level. Each vertex is assigned an index via a spatial hash function, which serves as a pointer to a learnable $d$-dimensional feature vector within a per-resolution look-up table.

At each level of the hierarchy, the feature vectors of the surrounding vertices are retrieved and combined via a trilinear interpolation based on the relative position of x within the local grid cell. The resulting interpolated vectors from all $L$ levels are concatenated to form a composite embedding $e(x) \in \mathbb{R}^{Ld}$. This multi-scale representation allows the network to simultaneously capture global structural context and local high-frequency variations, facilitating stable convergence and superior reconstruction of complex tissue boundaries.

### A.4. Framework Configuration

The framework is comprised of two coordinate-based MLPs utilizing SIREN (Sinusoidal Representation Networks) architectures (Sitzmann et al., 2020). Sinusoidal activation functions were selected to mitigate spectral bias and effectively capture the high-frequency components of the spatio-temporal ASL signal. The hyperparameters, specifically the first-layer frequency ($\omega_0$) and subsequent layer frequencies ($\omega$), were determined via grid search to maximize signal fidelity and the anatomical coherence of the resulting CBF and ATT maps.

To enforce physical plausibility, a non-negativity constraint was applied to the output layer of the parameter network ($f_{ode}$) using an exponential activation function. This ensures that predicted physiological values remain strictly positive, consistent with biophysical theory. Both MLPs were architecturally symmetric, utilizing 3 layers and 16 hidden units per layer.

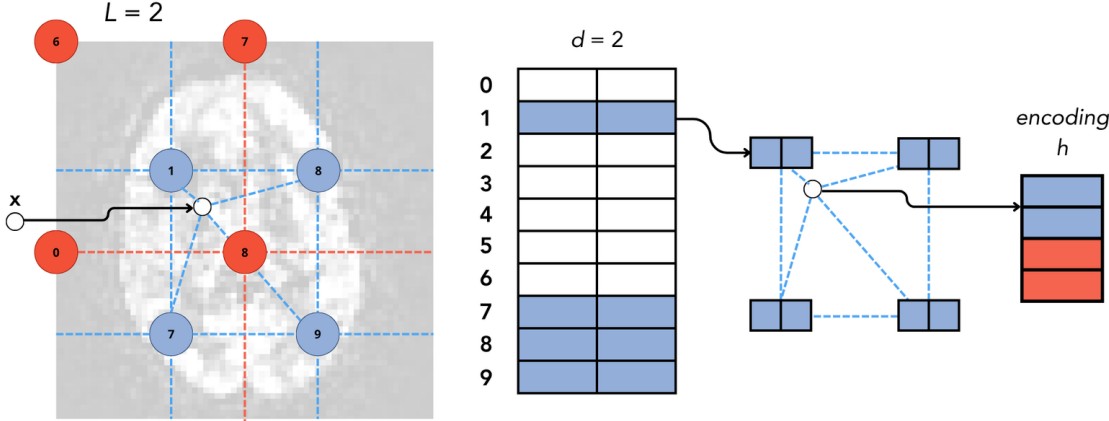

Figure 3: Multi-resolution hash-encoding in a 2D exemplary scenarios. Figure inspired by Müller et al. (2022) and de Vries et al. (2024)

Learning rate schedules, including initial and final values and cosine annealing period, were similarly optimized via grid search alongside the loss weights $(\alpha, \lambda)$.

The networks are trained jointly end-to-end using a multi-task objective. For each training batch, the first network ($f_{tissue}$) maps the spatio-temporal coordinates (t, x, y, z) to a predicted ASL signal for each one of the voxels in the batch. The data fidelity loss term ($L_{data}$) is computed as the Mean Squared Error (MSE) between the predicted signal and the observed signal at the acquired post-labeling delays (PLDs).

To regularize the model across the continuous temporal domain, we employ a collocation point strategy. For each voxel in the batch, the discrete acquisition time vector is augmented with four additional time points sampled from a uniform distribution between the first and the last PLDs. This new temporal variable is fed together with the spatial coordinates as input to the first network $f_{tissue}$, and the temporal derivative of the predicted signal is computed through backpropagation within respect of the time.

Simultaneously, the second network ($f_{ode}$) maps the spatial coordinates (x, y, z) to local CBF and ATT estimates. These estimates serve as inputs to the ODE of the Buxton biophysical model to compute an analytical derivative. To account for the time-varying nature of the labeling duration in the specific te-pCASL sequence, we modeled the bolus duration as a function of PLD using a fifth-order spline. The spline's derivative was integrated into the ODE term to ensure accurate physical grounding. The second loss term ($L_{phys}$) minimizes the discrepancy between the network-derived numerical derivative and the biophysically-

derived analytical derivative, ensuring the learned signal manifold adheres to the underlying perfusion dynamics.

Finally, the two losses are weighted and summed together, and the update of the parameters for both networks takes part. In this way, while the first network is forced to to follow not only the data but also the physics describing them, the second network provides physiologically coherent estimates of the parameters of interest.

