# OpenReview forum: "Prior-Free Arterial Spin Labeling Quantification via 3D Physics-Informed Neural Networks"
_MIDL.io/2026/Short_Papers — MIDL 2026 - Short Papers Poster_

### Official Review · Reviewer_rRCu · 2026-05-03
**This is an elegant PINN/INR formulation, but it has some unsupported claims**

**Rating:** 4
**Confidence:** 4

**Review:**

See strengths and weaknesses.

**Summary:**

The paper addresses parameter estimation in time-encoded pseudo-continuous
ASL (te-pCASL), where conventional Variational Bayesian estimators
 rely on explicit spatial priors to stabilize a low-SNR non-linear
inversion. The proposed approach is a per-subject 3D Physics-Informed
Neural Network with two SIREN-based MLPs: f_tissue(x,y,z,t) maps
spatio-temporal coordinates to the ASL signal, and f_ode(x,y,z) maps
spatial coordinates to CBF and ATT. Multi-resolution hash encoding is applied to spatial inputs to overcome MLP spectral
bias. The Buxton kinetic model is enforced as a differentiable physics
constraint inside a multi-task loss, so the time-derivative of f_tissue
must match the Buxton operator applied to f_ode. The framework was
evaluated on 18 healthy volunteers; the headline result is a 46% RMSE
reduction in gray matter and 37% in white matter relative to BASIL with
a Gaussian ATT prior (mean=1s, SD=1s), with PINN winning in 74% of GM
voxels and 76% of WM voxels.

**Strengths:**

1. The architectural choice is well-motivated and integrates several
  recent ideas appropriately: physics-informed loss for biophysical
  consistency, SIREN to mitigate spectral bias in coordinate networks,
  and multi-resolution hash encoding to capture fine anatomical
  structure across a 3D volume etc. The two-network factorization
  (f_tissue, f_ode) is intriguing to me, as it separates the manifold-fitting and the
  parameter-estimation roles.

2. The Buxton operator is integrated as a true differentiable
  constraint via automatic differentiation of f_tissue with respect to
  t, rather than just appearing as a regularizer on the output. This
  is conceptually closer to PINN literature and to me is the right approach.

**Weaknesses:**

1.  "Lower RMSE --> more accurate parameters" is the core inferential
  leap, and it is not fully justified. RMSE is computed between the measured
  ASL signal and the Buxton-model reconstruction using the *estimated*
  parameters. With no explicit spatial prior, a more flexible model
  (for example PINN with hash encoding has substantially more capacity than a
  voxel-wise VB) will trivially fit the noise better. VB intentionally
  trades fit for stability via its prior. The paper needs at least one
  of: (a) digital-phantom simulation with known CBF/ATT to demonstrate
  parameter accuracy, (b) test-retest reproducibility analysis, or
  (c) physiological plausibility checks (CBF in expected
  40–60 ml/100g/min range in GM, ATT in 0.7–1.5 s range, etc.). None
  is provided.

2. The "prior-free" claim is a little bit overstated. Multi-resolution hash encoding
  with finest resolution 4096, 16-unit SIREN MLPs, the choice of
  L = 16 levels, and the loss weights α=1, λ=0.5 collectively encode
  a strong implicit prior (smoothness across hash cells, spectral
  bias of SIREN, bandwidth of the encoding). The paper would be more
  accurate framed as "implicit-prior" or "data-driven prior" than
  "prior-free."

3. I'm concerned about the sample size and population. n=18 healthy volunteers is small for any
  claim about "robust 3D perfusion quantification." The lack of
  patients (stroke, dementia, low-CBF cases) is also a big gap. VB
  priors mostly bias things in low-SNR regions, which is precisely
  where the comparator was not stress-tested. te-pCASL with PLD = 49 ms
  and total labeling 3600 ms is one specific protocol; generalization
  to other te-pCASL or single-PLD pCASL is not addressed.

**Justification Of Rating:**

This paper clearly is above the bar of a short paper track given the very limited # of pages. The method design is clever and well motivated.  Additionally, the paper tells an internally coherent story overall. That said, I do want to comment on one part. I found the story somehow breaks is at the inferential step from "lower RMSE on Buxton-reconstructed signal" to
"more accurate physiological parameters," and at the
"prior-free" framing. It is not fundamentally wrong, but the paper somewhat conflates "fits the measurements better" with "is right,"
and whose conclusion ("more accurate and anatomically consistent
perfusion maps") overstates what the experimental design
can support. For a MIDL short paper, telling a coherent and self consistent story is more important than proposing a "breakthrough" method. Should the paper be accepted by PC, the authors are encouraged to revise the claims in the final version.

---

### Decision · Program_Chairs · 2026-05-08

Accept (Poster)